# Advances in Rice Seed Shattering

**DOI:** 10.3390/ijms24108889

**Published:** 2023-05-17

**Authors:** Hao Wu, Qi He, Quan Wang

**Affiliations:** 1Shenzhen Branch, Guangdong Laboratory of Lingnan Modern Agriculture, Genome Analysis Laboratory of the Ministry of Agriculture and Rural Affairs, Agricultural Genomics Institute at Shenzhen, Chinese Academy of Agricultural Sciences, Shenzhen 518120, China; wuhao@caas.cn (H.W.); heqi@caas.cn (Q.H.); 2College of Agricultural Sciences, Nankai University, Tianjin 300071, China

**Keywords:** rice, seed shattering, abscission layer, regulatory network, domestication

## Abstract

Seed shattering is an important trait that wild rice uses to adapt to the natural environment and maintain population reproduction, and weedy rice also uses it to compete with the rice crop. The loss of shattering is a key event in rice domestication. The degree of shattering is not only one of the main reasons for rice yield reduction but also affects its adaptability to modern mechanical harvesting methods. Therefore, it is important to cultivate rice varieties with a moderate shattering degree. In this paper, the research progress on rice seed shattering in recent years is reviewed, including the physiological basis, morphological and anatomical characteristics of rice seed shattering, inheritance and QTL/gene mapping of rice seed shattering, the molecular mechanism regulating rice seed shattering, the application of seed-shattering genes, and the relationship between seed-shattering genes and domestication.

## 1. Introduction

Rice is one of the world’s most important crops. Over 90% of the world’s rice is produced and consumed in Asia alone [1,2]. According to archaeological research in the middle and lower reaches of the Yangtze River, the currently cultivated rice (*Oryza sativa* L.) in Asia was artificially domesticated from the common wild rice (*Oryza rufipogon* G.) about 10,000 years ago [3]. During domestication, changes in seed shape, decreased dormancy, increased seed numbers, reduced seed shattering, and improved plant type and fertility are all important events in artificial selection [4,5,6,7,8]. Compared with wild rice, the loss of seed shattering in cultivated rice is one of the major trait changes and an important epitome of the transition from wild rice to cultivated rice. Reducing seed shattering made rice easier to harvest intensively, greatly improving the yield and significantly contributing to human reproduction [9,10].

In *Oryza*, the seed-shattering ability of different varieties varies greatly. Generally, *indica*-type rice (*Oryza sativa* L. subsp. *indica* Kato) seeds are easy to abscise, and *japonica*-type rice (*Oryza sativa* L. subsp. *japonica* Kato) seeds are difficult to shed from the pedicel [11]. During normal harvests, *indica* rice and some easy-shattering *japonica* rice cause yield losses due to seed shattering. The yield loss is irreparable when severe weather occurs during rice growth or seed maturity. Moderate-shattering rice varieties are suitable for large combined harvesters or manual harvesting and threshing, and hard-shattering rice varieties are suitable for small semi-feeding harvesters. Different harvesting methods for different seed-shattering degrees of rice varieties can significantly improve harvest efficiency and reduce energy consumption [12,13,14]. Therefore, understanding the molecular mechanism of rice seed shattering is significant for cultivating rice varieties with moderate seed-shattering degrees, reducing yield loss caused by seed shattering, and improving harvest efficiency.

## 2. Physiological Basis of Seed Shattering in Rice

Rice seed shattering is a complex agronomic trait, that is affected not only by environmental factors such as light, temperature, humidity, pests, and diseases but also by its genetic factors such as panicle shape, awn trait, and formation and fracture of the abscission layer (AL), also known as the abscission zone (AZ) (Figure 1). One of the major determinants is AL formation, which consists of several layers of small, dense cytoplasm and flat parenchyma cells at the junction of sterile lemma (sl) (also known as glume or empty glume) [15,16,17] and rudimentary glume (rg), while the surrounding pedicel and glume cells consist of large sclerenchyma cells. There are four stages from AL formation to seed shattering. First, the differentiation of the AL: the formation of the AL in rice occurs 16–20 d before heading. Second, the development of the AL’s competence in responding to abscission signals: plant hormones are the most important regulatory signals, and ethylene, abscisic acid, and jasmonic acid promote the dissociation process, while auxin and brassinolide inhibit it. Third, cell wall modification and cell separation: due to the influence of specific signals and environmental factors, the activity of cellulase, polygalacturonase, β-endoglucanase, and other hydrolases in the AL cells is initiated, and then the degradation of the middle lamella and cell wall of the AL cells leads to the seed separation from the plant. Fourth, the establishment of a protective layer: a protective layer is formed on the separation surface of the AL [18,19,20,21,22].

Different morphological characteristics of the AL are the fundamental reason for the difference in the seed-shattering degree of rice varieties. A well-developed AL extends from the epidermis to the vascular bundle in wild rice. Mature grains are easily shed, which continues the rice species and is conducive to the spontaneous growth of wild rice. Additionally, in easy-shattering rice varieties, a well-developed AL is also formed. However, some varieties have an upward-sloping AL, from the epidermis to the vascular bundle, with a wider support zone (sz) of non-degraded parts between the two ALs on both sides, resulting in moderate to hard-to-shed traits. Among cultivated rice varieties, *japonica* rice shows a delayed AL emergence compared to *indica* rice, and no AL is formed, even after seed maturation [23]. Moreover, the number and size of the AL cells are small, so *japonica* rice is more difficult to shed than *indica* rice. Additionally, the broken interface of the AL in the spikelet basal part, examined by scanning electron microscopy (SEM), shows the non-shattering varieties display a rough and broken cross-section with spring-like broken vascular bundles. Whereas there is a smooth fracture surface and a clear circular border in easy-shattering rice varieties [24]. However, the seeds of non-shattering rice varieties also fall off from pedicels when subjected to external forces, some breaking off at the same position as the easy-shattering varieties. However, at present, research on rice varieties without an AL is not detailed enough. For example, why are the AL developmental genes of the non-shattering rice varieties suppressed, resulting in ALs not forming?

## 3. Factors Affecting Seed Shattering

Many environmental factors can lead to rice shattering, such as photoperiod, water stress, wounding and pathogen attack, and other biotic or abiotic stresses [25,26,27,28,29]. The quality (wavelength) and quantity (intensity/duration) of light are critical to photomorphogenesis, so plants have developed intricate light detection systems involving multiple photoreceptors that can regulate diverse developmental processes including organ abscission [30,31,32]. Additionally, salt, cold, and high temperatures cause a water deficiency, which can also promote abscission due to a decline in the growth and vigor of the plant [33,34,35]. Injuries caused by herbivore feeding or other mechanical damage can provide a possible entry point for pathogens that induce a plant defense response. Additionally, infected organs will be shed first to prevent the spread of infection throughout the affected plant and its neighbors. Most of these factors that affect shattering are essentially due to phytohormone changes aside from some direct mechanical damage, such as gales and rainstorms, and gnawing by animals or birds. Plant hormones play an important role in AL formation, and rice shattering is the result of the synergistic and balanced effects of various hormones [36]. As mentioned in Section 2, ethylene is an important hormone that ensures the timely shedding of grains, while auxin inhibits the shedding [37,38]. The shedding process also involves some hydrolases related to cell wall degradation, including beta-1,4-glucanase and polygalacturonase. Hydrolase activity is activated in AL cells, resulting in the degradation of the intercellular layer and cell wall. Moreover, expansin, pathogenesis-related proteins, and several other proteins also play important roles in seed-shattering progress [18].

## 4. Seed-Shattering QTL Mapping and Gene Cloning in Rice

The common methods for identifying the shattering degree are rubbing, falling, and instrument measurement. The rubbing method refers to holding the rice panicle with one hand, gently rubbing, and observing the seed shattering. This method is the most simple and efficient, but its results are easily affected by humans, and it is only suitable for the mapping of quality trait genes and major QTLs in large populations [39]. The falling method entails letting the rice panicle fall freely from a certain height to an iron plate laid on the ground and taking the percentage of the number of seeds shattering from the total number of seeds from the panicle as the falling rate [40]. The instrument measurement method is to determine the force that causes the seed to fall from the branch using a digital dynamometer, and judge the difficulty of seed shattering according to the force [41]. The measurement results of the instrumental method are accurate and can avoid experimental errors caused by humans. However, the operation is time consuming and laborious, and it is only suitable for identifying multi-gene-controlled traits or micro-effect QTLs.

At present, the prevailing view about seed-shattering inheritance is that it is a quantitative trait, that may be controlled by multiple genes, or by a few major genes and multiple minor genes. Up to 2020, at least 60 QTLs/genes related to rice shattering had been detected in 12 rice chromosomes, using various genetic populations and linkage maps (Table 1). The vast majority of QTLs are identified by crossing between *japonica* rice, *indica* rice, wild rice (*Oryza rufipogon*), and weedy rice. Very few QTLs are identified by crossing between *javanica* and *india*. These different QTLs reflect the diversity of shattering phenotypes both intra-species and inter-species. Some QTLs have not been explored in depth because they have weak effects or are located in the same chromosome region, which may be alleles to known genes. For instance, the main QTL effect is detected in the first and fourth chromosomes in many genetic populations, which may be alleles to *qSH1* (*a QTL of seed shattering on chromosome 1*) and *sh4* (a *seed-shattering-related QTL on chromosome 4*) from Asian cultivated rice (*Oryza sativa*) and wild rice (*Oryza rufipogon* Griff), respectively. Additionally, the frequency of QTLs detected on chromosomes 1, 3, and 4 is higher. These shattering-controlling QTLs can be detected in wild rice (annual or perennial), weedy rice, and cultivated rice. Thus, it is important to accurately locate and clone related genes based on these QTLs to elucidate the regulatory network of seed shattering. A total of 12 genes related to seed shattering have been cloned up to now (Table 2), among which, *qSH1*, *SH4*, *Sh1/ObSH3*, *SHAT1*, *SH5*, *CPL1*, *SNB,* and *ObSH11* are the major genes controlling AL formation and seed shattering. *OSH15*, *GRF4*/*PT2*, *NPC1*, and *LG1* are pleiotropic genes that have significant effects on other agronomic traits. Most of these genes are transcription factors, such as MYB transcription factor, *SH4*, and *OgSH11*, BEL-1 type homeobox transcription factor *qSH1*, and KNOX protein SH5, AP2 transcription factor, *SNB* and *SHAT1*, and the YABBY transcription factor, *Sh1/ObSH3*. Additionally, some of them encode kinases, such as *CPL1* and *NPC1* [42,43,44,45,46,47,48].

Mutations of an MYB transcription factor *SH4* lead to a non-shattering phenotype of both Asian (*Oryza sativa*) and African (*Oryza glaberrima*) cultivated rice, which are domesticated from two wild rice species, *Oryza rufipogon* [64] and *Oryza barthii* [4,76,77], respectively. A mutation (G/T) at base 237 on the first exon of the *SH4* allele in cultivated Asian rice results in the conversion of aspartic acid to serine, leading to incomplete AL development. Lin et al. [13] located and cloned *SHA1* (*Shattering1*), encoding a plant-specific trihelix transcription factor family, using a backcrossed population between cultivated rice Teqing and an introgression line IL105 with the seed-shattering habit derived from perennial common wild rice, YJCWR. The amino acid sequences of SHA1 and SH4 are 98% consistent, and the mutation of G to T is also found at base 237 in the coding sequence. However, the histological analysis and gene transformation experiments showed SHA1 was not involved in AL formation but was related to the AL cells’ degradation. Compared to *SH4*, *SHA1* has an extra 6 bp insertion at 158 to 163, which may lead to different functions for these two genes. Although the downstream target gene of *SH4* is unknown, it may be involved in programmed cell death or hydrolytic enzyme release. Interestingly, the non-shattering *SH4* allele has also been found in some *O. Nivara* wild rice, probably due to the introduction of the non-shattering allele into wild germplasm through hybridization with domesticated plants.

A SNP (single-nucleotide polymorphism) in the 5′ regulatory region of *qSH1*, a BEL1-type homeobox gene, which is a rice orthologue of *RPL* (*REPLUMLESS*) in *Arabidopsis thaliana*, causes loss of seed shattering owing to the absence of AL formation. This SNP explains 68.6% of the shattering difference between *indica* and temperate *japonica* cultivars [24]. Another BEL1-type homeobox gene, *SH5*, is highly homologous to *qSH1*. AL development and seed loss are reduced when *SH5* is suppressed in easy-shattering Kasalath. Additionally, overexpression of *SH5* in moderate-shattering Dongjin and non-shattering Ilpum leads to an increase in seed shattering, because lignin levels are decreased in the basal region of the spikelet [69].

Zhou et al. [66] used the *indica* rice variety GLA4 and common wild rice W1943 to construct an introgression line SL4 containing the cloned genes *SH4* and *qSH1*. By using radiation mutagenic SL4, two non-shattering mutants, *shat1* and *shat2*, were isolated and identified even when *SH4* and *qSH1* existed together. Among them, loss of shattering in *shat1* is controlled by a gene *SHAT1* (*shattering abortion1*) that encodes the AP2 transcription factor, and *shat2* is caused by a mutation in the known shattering gene *SH4*. A 1-bp deletion in the first exon between the nucleotide sites +41 and +42 in the *shat1* mutant, resulted in a frameshift that led to a non-shattering phenotype. Jiang et al. [71] used YIL100, an introgression wild rice line with a strong shattering character, to construct an EMS mutant library and screen out a mutant *ssh1* (*suppression of shattering1*) with a reduced shattering degree. In combination with MutMap mapping and a genetic transformation experiment, *SSH1*, an allele of *SNB* (*SUPERNUMERARY BRACT*), encoding the plant-specific AP2 transcription factor that controls shattering, was identified. A C-to-A point mutation in the ninth intron of SNB alters the splicing of its messenger RNA, causing reduced shattering by altering AL development. The introgression of *SNB* mutant allele *ssh1* into *indica* rice variety 93-11 can increase grain length by 9.5% and 1000-grain weight by 7.7%, indicating *ssh1* can improve the rice yield.

*OsSh1* (*Shattering1*) encodes the YABBY transcription factor and is a homologous gene of *sh1* that controls seed shattering in sorghum. Using the rice non-shattering mutant (SR-5) and the wild-type rice breeding line (Nanjing 11), Lin et al. [67] conducted genome-wide comparison and transcriptional expression analysis; insertion of a >4 kb fragment was identified in intron 3 of *OsSh1*, leading to reduced levels of transcription and the shattering-resistant phenotype. *ObSH3* (*Oryza barthii seed shattering 3*), derived from wild African rice, is located on chromosome 3 and encodes a YABBY transcription factor. The deletion of the *ObSH3* genome segment in African cultivated rice leads to asymmetrical and incomplete AL development, resulting in shattering loss [68]. Ishikawa et al. [23] identified a causal single-nucleotide polymorphism at *qSH3* within the seed-shattering gene *OsSh1*, which is conserved in *indica* and *japonica* subspecies but absent in the *circum-aus* group of rice. They showed that *SH4* alone is insufficient to reduce shattering, and *qSH3* is required to cause AL disruption.

Ji et al. [41] characterized a shattering mutant line of rice, *Hsh*, derived from a non-shattering *japonica* variety, Hwacheong. It was concluded that the easy shattering of *Hsh* is controlled by the single recessive gene *sh-h*, which encodes a protein containing a conserved carboxy-terminal domain (CTD) phosphatase domain, named *OsCPL1* (*Oryza sativa CTD phosphatase-like1*). *OsCPL1* represses differentiation of the AL during panicle development. Cao et al. [73] indicated that *NPC1* (*non-specific phospholipase C1*) modulates silicon distribution and secondary cell wall deposition in nodes and grains, affecting mechanical strength and seed shattering. *NPC1* overexpression lines have brittle stem and panicle nodes that are snapped easily by bending, which leads to the mature seeds being easily threshed off the head, increasing seed shattering.

*OSH15* (*Oryza sativa homeobox 15*) encodes a knotted-type homeodomain protein. Previous studies indicated *OSH15* was expressed in a region where the shoot apical meristem would develop later in early embryogenesis and in a ring-shaped pattern at the boundaries of some embryonic organs in late embryogenesis [78]. Yoon et al. [79] demonstrated that *OSH15* mRNAs were abundant in AL during spikelet development. Additionally, OSH15 and SH5 interact directly with *CAD2* (*cinnamyl alcohol dehydrogenase 2*) chromatin to inhibit lignin content, leading to a reduced seed-shattering phenotype. *GRF4* (*Growth-Regulating Factor 4*) encodes a growth-regulating factor that positively regulates grain shape and panicle length and negatively regulates seed shattering [74,80]. Ishii et al. [75] produced two backcross populations with reciprocal genetic backgrounds: cultivated rice Nipponbare with closed panicles and wild rice W630 with spreading panicles. The *SPR3* (*Spreading Panicle3*) locus is identified as a 9.3 kb genomic region, and complementation tests suggest this region regulates *OsLG1* (*liguleless gene 1*). A mutation at *SPR3* in *O. rufipogon* changes the panicle structure from open to closed, which leads to a reduced seed-shattering morphology. Recently, Ning et al. [72] demonstrated that African cultivated rice showed significantly reduced seed shattering by knockout of *SH11*, because *OgSH11* represses the expression of lignin biosynthesis genes and lignin deposition by binding to the promoter of *GH2* (*gold hull and internode-2*)/*CAD2*. However, this gene, called *JAMyb*/*MYb21* in *Oryza sativa*, is associated with biotic and abiotic stresses [81,82]. No related research has been reported previously regarding rice seed shattering in this gene.

## 5. Regulatory Network of the Seed-Shattering Gene in Rice

An increasing number of genes related to AL formation or seed shattering have been identified up to now, but it is not as thorough as the research in the dicot model plant *Arabidopsis thaliana* [83,84,85,86,87,88,89,90], and the regulatory network specifying AL development in rice has remained obscure [91,92]. Some researchers have conducted constructive explorations into establishing a regulatory network among seed-shattering genes. Additionally, an evolutionary model of AL formation has been proposed based on these studies (Figure 2) [66,91]. However, it must be emphasized that these genes affect shattering through different mechanisms and come from distinct varieties or species with diverse genetic backgrounds. It is difficult to directly identify the shattering genes’ regulatory relationships from a genetic perspective. This figure is just a collection of existing results without considering the genetic background.

According to Itoh et al. [15], rice inflorescence and spikelet development are divided into nine stages (In1–In9) and eight stages (sp1–sp8), respectively. Additionally, some of the developmental stages overlap. Current data suggest *SH4* and *qSH1* play crucial roles in the formation of AZ and seed shattering, while *qSH1* is epistatic to *SH4* [24,63,64,93]. According to in situ hybridization in *japonica* cv Nipponbare, *SH4* is the first expressed gene in AL among these seed-shattering-related genes. *SH4* expression emerges in sp6 (spikelet development stage 6) and remains from sp7 to sp8e (early stage sp8) but disappears during sp8l (late stage sp8) [66]. *qSH1* expression is detected at the provisional AL position only in *NIL* (*qSH1*) and not in Nipponbare in ln7 (inflorescence development stage 7) [24]. Zhou et al. [66] also demonstrated that no *qSH1* signals were detected in Nipponbare, but strong signals were detected in *Oryza rufipogon* W1943 in Sp8e. Additionally, *SHAT1* expression persists in AL from sp8e to sp8l. However, the *qSH1* and *SHAT1* expression in AL is abolished in the *sh4* mutant, which suggests *SH4* acts upstream of *qSH1* and *SHAT1*. Moreover, the inability to detect *qSH1* in the *shat1* mutant suggests that *qSH1* is activated by *SHAT1*. Conversely, the defective AZ phenotype in Nipponbare is suppressed by sustained expression of *SHAT1* and *SH4* in the AZ after introgression of a functional *qSH1* locus from *indica* cv 93-11. These results suggest that *qSH1* maintains *SHAT1* and *SH4* expression in the AZ [66].

*SH5* is a homologous gene to *qSH1,* with 70% identity and 77% similarity. Similarly, the *SH5* gene is expressed at an early stage in sp7 of the abscission process and positively affects the expression level of *SH4* and *SHAT1* in the *SH5* activation tagging line, according to qRT-PCR and in situ hybridization [69]. OSH15 is expressed preferentially in the AZ during sp7 and early in sp8 and OSH15 interacts with qSH1 and SH5. In addition, OSH15 can interact with SH5 and qSH1 in the CO-IP experiment, which together enhance the development of the AZ [79,94]. Moreover, SNB positively regulates the expression of *qSH1* and *SH5* by binding to their promoters directly, thus controlling seed shattering [71]. Moreover, *OsLG1* and *OsGRF4* regulate shattering by affecting panicle architecture, and plants with spreading panicles are easier to shatter. *OsNPC1* affects stem mechanical strength and seed shattering by mediating the distribution of silicon in stem nodes, grains, and secondary cell walls. *OgSH11* alters the lignin content in the base of the spikelet to affect its shattering ability. All of the other genes regulate shattering by influencing the formation and development of the AL.

## 6. Seed Shattering and Domestication

Humans began to shift from hunting and gathering to an agricultural society around the world about ten thousand years ago. Our ancestors bred high-yielding crops from hundreds of wild plant species that humans rely on today, such as rice, wheat, corn, millet, and so on. About four thousand years ago, all major crop varieties completed the domestication process [95]. The main domestication traits of crops can be summarized into the following six characteristics: 1. Loss or reduction in natural dispersibility of seeds; 2. reduction in self-propagation characteristics of seeds; 3. increase in seed or fruit size; 4. lack of germination inhibition mechanism; 5. change from perennial to annual; 6. compact growth habits [96]. In domesticated rice, over 20 agronomic traits are identified as the comprehensive domestication characteristics of rice (*Oryza sativa)* [97]. As with other cereal crops, the reduction in seed shattering is one of the most significant changes during the domestication process. Rice seeds’ non-shattering trait is caused by genetic mutations under unconscious and conscious selection pressure after artificial cultivation. Isolation and identification of key domesticated genes of seed dispersal and the analysis of their molecular evolution could provide new insights into the crop’s origins and evolution.

Among the seed-shattering genes that have been identified, some have been artificially selected. *SH4* expression slowly increasing in *O. sativa* less than in *O. nivara* during grain maturation might have been a result of selection in the regulatory region of the gene to keep the shattering/threshing balance during rice cultivation [64]. In African rice domestication, an SNP resulting in a truncated SH3/SH4 protein in the cultivated rice species *Oryza glaberrima* caused a loss in seed shattering compared to the wild species *Oryza barthii* [98]. However, the *SH4* mutation site in Asian cultivated rice (G/T) is different from African cultivated rice (C/T), indicating that the two genes are selected in parallel and the two cultivated species are independently domesticated. However, this does not lead to the complete loss of *SH4* gene function in cultivated rice; that is, the mutation of *SH4* in cultivated rice does not lead to the complete disappearance of the AL but only leads to a decrease in the degree of seed shattering in cultivated rice. In this way, human beings have enough time to harvest rice after maturity, which is easy to harvest, to guarantee the grain yield. Zhang et al. [63] demonstrated that the *SH4* non-shattering alleles were fixed in *indica* and *japonica* subspecies through long-term artificial selection. Molecular evolutionary analysis of *SH4* shows U.S. weedy rice originates from cultivated rice but re-acquired the shattering trait after divergence from its progenitors through alternative genetic mechanisms [99]. Lv et al. [68] investigated the genetic relationship of 93 *O. glaberrima* and 94 *O. barthii* accessions using previously published resequencing data [76,77], and indicated the geographical distribution of the *SH4* gene was more extensive, suggesting *SH4* may have been selected for a longer time in African cultivated rice. Additionally, system reconstruction shows *SH4* in *japonica* and *indica* are on the same clade, and their polymorphism is significantly reduced. Neutrality tests and coalescent simulation tests also show the *SH4* gene is retained on the genome through single domestication, which is considered to be the strongest evidence for a single origin of rice or the emergence of *indica*/*japonica* differentiation after domestication [13,100].

Konishi et al. [24] demonstrated, in the hypothetical process of the evolution of *qSH1*, that the SNP distribution (G in Kasalath and T in Nipponbare) revealed a strong selection by ancient humans for the SNP in early domesticates of *japonica* subspecies. Haplotype analysis shows all easy-shattering rice varieties have the same SNP locus as Kasalath, while hard-shattering varieties have the same SNP locus as *japonica*. However, the selection of *qSH1* is not obvious in *indica* rice and not clear in *japonica* rice [63]. Phylogenetic analysis shows *qSH1* is distributed in different clades of the two subspecies, indicating that *qSH1* has different domestication histories in the two subspecies [7]. Additionally, breaking tensile strength measurements in some temperate *japonica* rice indicates this allele is associated with the level of shattering, suggesting selection for the *qSH1* allele is not as strong and widespread as selection for *SH4* since *qSH1* is not fixed even in temperate *japonica* rice, let alone in the entire cultivar. Since *qSH1* is identified in separate populations of *indica* and *japonica* hybrids, Konishi et al. [24] suggested that the loss of seed shattering in both *indica* and *japonica* subpopulations was an independent genetic change during the process of artificial domestication.

Furthermore, in two whole-genome sequencing studies, *OsSh1* is on a list of genes shown to experience strong artificial selection [67,101,102]. Jiang et al. [71] found the level of sequence polymorphism at the 5′-flanking regions of SNB was strongly reduced in both *indica* and *japonica* varieties relative to the wild progenitors, which indicated the natural variations in SNB might be associated with domestication. Zhou et al. [66] showed *SHAT1* might not be subject to artificial selection during domestication according to high-throughput sequencing results of 614 accessions of landraces from China and 330 accessions of international varieties, which showed 19 SNPs located in the *SHAT1* genic region, which generated few functional variants [103]. There is no evidence that other abscission genes were artificially selected during domestication.

As with rice, reducing seed shattering in other cereal crops (*Poaceae*) was one of the most significant changes during domestication, indicating artificial selection also acted on them to reduce yield loss caused by seed shattering. Most of the cloned rice-shattering-related genes encode transcription factors and are relatively conserved. Homologous genes from different species associated with shattering traits often have the same or similar biological functions. In wheat (*Triticum aestivum*), the *Q* gene is related to the shape and toughness of wheat glume and is an important shattering domestication gene. The *Q* gene encodes an AP2 transcription factor, which has a high amino acid similarity to the rice *SNB* gene, and its q allele is characterized by an elongated spikelet and non-shattering in wild wheat [104]. *TaqSH1-D* is identified in the F2 generation mapping population of wheat, which is homologous to the rice shattering gene *qSH1* [105,106]. It affects rachis degradation, which leads to an easy-shattering phenotype when wild barley ripens [107]. Lin et al. [67] cloned the first *Sh1* gene from the F2 population of wild sorghum (*Sorghum propinquum*), which encodes the YABBY transcription factor and belongs to the YABBY family with rice *OsSh1* and maize (*Zea mays*) *ZmSh1-1* genes, which are lineal homologous genes. The mutation of *Sh1* inhibits its expression and changes the protein, thus reducing its shattering ability. This research suggests convergent domestication may play a significant role in cereal. The same adjustable single-nucleotide polymorphism is found among different crops, which may be related to the developmental variation in the seed dispersal structure associated with domestication and the natural selection of distant species. These studies provide a reference for cultivating shattering-resistant crops via the parallel selection of the same potential genetic targets [108,109]. Therefore, rapid cloning of genes related to shattering by reverse genetics will provide a new way for the study of shattering characteristics. This study can also be extended to the research on shattering traits of other non-model crops to solve the current practical problems in the agricultural production of non-model crops and deepen our understanding of crop domestication.

## 7. The Application of Seed-Shattering Genes

The degree of shattering has an important effect on the rice yield and harvesting method. Breeding rice varieties with a moderate shattering degree benefits efficient harvesting and avoids yield loss. Currently, *indica* hybrid rice is the main crop used in commercial rice production, especially in southern China. However, compared to *japonica* rice varieties, these *indica* varieties show a greater tendency toward “easy-shattering” traits. Previous reports indicated certain hybrid rice varieties suffered 5.8–8.6% yield losses due to seed shattering [110]. Therefore, the development of new rice cultivars with intermediate shattering phenotypes has been a priority in recent years. Based on previous research, there are three main strategies to use the seed-shattering genes in rice breeding to obtain varieties with moderate seed-shattering degrees. 1. CRISPR/Cas9 technology (clustered regularly interspaced short palindromic repeats/CRISPR-associated 9) has been widely applied as a means of rapidly and reliably conducting genomic editing in rice. Therefore, many research projects have used CRISPR/Cas9 to edit rice seed-shattering genes to change shattering degrees in functional rice breeding [111,112,113,114]. Sheng et al. [115] conducted targeted mutagenesis of the *qSH1* gene via CRISPR/Cas9 in the *indica* hybrid rice cultivar GLY1128 (GuangLiangYou1128), which has excellent agronomic traits but a strong seed-shattering phenotype. Using the gene editing approach, the shattering degree of *qsh1* mutants was significantly reduced and there were no significant changes in other agronomic traits compared with wild-type plants under normal growth conditions. Furthermore, de novo domestication of wild plants as a new crop breeding strategy was proposed [116,117]. Allotetraploid wild rice has the advantages of large biomass, heterosis, and strong environmental adaptability, but it also has non-domestication traits, such as sparse spikelets, small grain size, a long awn, and seed shattering easily, and cannot be used for modern agricultural production. Several domestication-related genes, including the abscission gene *qSH1* and panicle architecture gene *LG1*, have been selected for gene editing experiments and successfully created various materials with a lower shattering degree, shorter awn length, lower plant height, longer grain size, thicker stem, and shorter heading time. It is possible that de novo domesticated allotetraploid rice could be developed into a new grain to enhance world food security. 2. Using γ-ray mutagenesis technology to select moderate seed-shattering rice varieties. To improve the easy shattering trait of fine super rice ‘Xieqingzao A/T9308’ and prolong the production life, the easy shattering restorer line T9308 was mutated by γ-ray mutation, a mutant M9308, with its main agronomic traits unchanged but seeds that were significantly difficult to shed, was screened, and a new super rice combination ‘Xieqingzao A/M9308’ with a medium shattering habit was selected for mating. A field investigation found ‘Xieqingzao A/M9308’ could significantly reduce the rice loss rate in nature and machine cutting for sustainable application in production [118]. 3. Using natural or artificial mutants of the seed-shattering gene to generate gene introgression lines to modify the shattering ability. As mentioned above in Section 4, the introgression of the EMS-generated *ssh1* allele, an *SNB* gene mutant, into the current cultivar 9311, generated 9311*^ssh1^* which had increased seed lengths and 1000-grain weights compared with the 9311*^SSH1^* plants [71].

The seed-shattering gene is not only useful in rice breeding but also plays a very important role in managing the infestation of wild rice relatives (weedy rice, *Oryza sativa* f. *spontanea,* and weedy *Brassica* taxa), which may result in unwanted environmental consequences [119,120,121]. Briefly, seed-shattering genes can be knocked out in cultivated rice through a gene silencing technique to form a “safe box” that contains seed-shattering gene silencing components. Then, the component may be transferred into weedy rice through pollen-mediated gene flow (transgene flow) from cultivated rice, reducing or weakening the seed-shattering ability of weedy rice. Furthermore, it is important that silencing seed-shattering genes has little or no adverse effect on cultivated rice. Yan et al. [122] demonstrated that partially silenced *SH4* expression in crop–weed hybrid descendants led to reduced seed shattering with no differences in productivity-related traits. Zhang et al. [123] edited *qSH1* and *SH4* in a weedy rice line (“C9”) with a strong seed-shattering degree; all of the gene-edited weedy rice lines showed a reduced seed-shattering phenotype without a consistent reduction in seed size. These results prove that it is feasible to reduce the shattering ability of weedy rice by gene-editing shattering genes. Once seed-shattering gene silencing components in the cultivated rice transgenic lines are transferred to weedy rice through transgene flow, the risk of transgene diffusion will be reduced due to the decreased seed shattering of weedy rice. At the same time, the seed shattering and dispersibility of weedy rice containing seed-shattering gene silencing components will be greatly reduced, to control the dispersal of weedy rice. In general, this approach can greatly reduce the environmental safety risk caused by transgenic escapes from weedy rice, reduce the spread and harm of weedy rice, and avoid economic loss.

## 8. Conclusions and Future Directions

Rice is one of the most important crops in the world and is identified as a monocotyledon model plant. The cloning of abscission genes can not only reveal the development and evolution of rice but also be used for genetic improvement. Because of the conservation of genes controlling seed-shattering traits during domestication in different species, it may also be applied to improving other *gramineous* crops. However, the process of gene regulation in seed shattering is complex, and the correlation between seed-shattering genes has not been deeply studied. Additionally, the function of seed-shattering genes is not completely conserved, some genes have different functions in *gramineous* species and interspecies, so the location of seed-shattering genes and the mechanism of gene interactions in *gramineous* species need to be further explored. 

With the development of molecular biology and genomics, it is becoming easier and faster to study domestication-related traits and elucidate the regulatory mechanisms of rice. Moreover, a better understanding of the molecular basis of rice seed shattering is of positive significance for modern breeding, because modern breeding can also be regarded as another “artificial domestication” of rice, namely “super-domestication” [5]. The mapping and cloning of seed-shattering genes have accumulated new materials for understanding the artificial domestication of rice and provided gene resources for breeding applications. Although many QTLs have been identified to control shattering, only a limited number of genes have been cloned, and how other genes affect shattering remains unclear. Additionally, some of the seed-shattering genes have been shown to undergo artificial selection during domestication, but the current research on rice domestication usually focuses on single genes or some neutral markers, but the evolution of the regulatory network of rice domestication traits is not studied as a whole. Compared with only considering the major genes of a single domestication trait, studying the process of rice domestication based on the network regulation pathway of domestication traits will more accurately reveal the evolution of cultivated rice under artificial selection. Fortunately, with the improvement of the quality of the reference genome of rice and the further improvement of genome annotation, whole-genome resequencing has been applied to the research of various rice varieties, and the molecular mechanism of domestication traits is becoming more and more clear, which brings the possibility to study the evolution of the gene regulatory network of domestication traits as a whole.

Moreover, rice seed shattering is a complex, multifaceted regulatory process. Hormones, enzymes, nutrients, and environmental factors affect seed shattering, but the mechanisms by which these factors interact to affect abscission have not been clarified. Studies on hormones were carried out earlier, such as the inhibitory effect of auxin on plant organ shedding and the promoting effect of ethylene. However, the regulatory effects or regulatory pathways between different hormones on abscission are still unclear. For example, the correlation between jasmonic acid and ethylene, ethylene, and abscisic acid on abscission is not quite clear. In addition, due to the variety of hormones, the regulation mechanism of some other hormones on seed shattering and the corresponding signal transduction mechanism also need to be further studied. For instance, gibberellin is an important hormone for plant growth and development throughout the plant’s whole life, including seed germination, stem elongation, floral transition, and fruit development, but whether gibberellin affects seed shattering and by what mechanism is still obscure [124,125,126,127]. The mechanism of enzymes in rice seed shattering was studied later than that of hormones. Therefore, people’s understanding of the physiological, biochemical, and molecular biology aspects of related enzymes in the abscission process needs to be further improved. In addition to strengthening the research on the synergistic effect of enzymes on plant shedding, it is also necessary to enhance the physiological mechanisms of related enzymes regulating the synthesis, degradation, transportation, and localization of plant organs during the abscission process. Therefore, the complex mechanism of rice seed shattering will need to be studied and explored for a long time. Additionally, cloning shattering genes, elucidating the molecular mechanism of shattering, and expounding the interaction network between shattering genes will remain the main research directions in rice seed shattering.

## Figures and Tables

**Figure 1 ijms-24-08889-f001:**
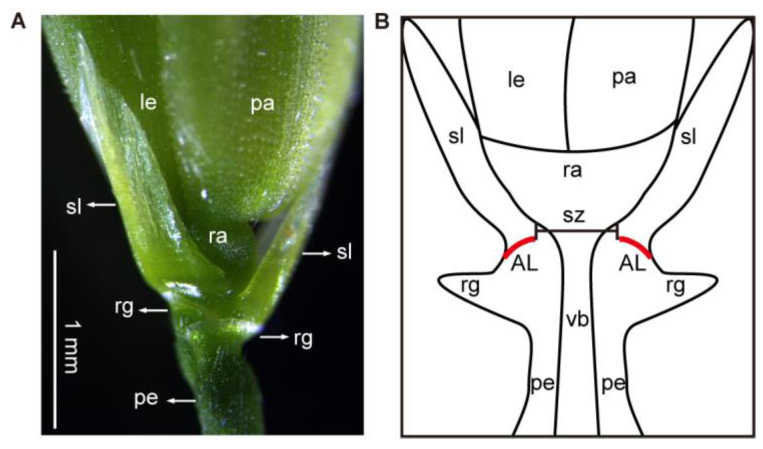
Structure diagram of rice spikelet base and AL. Photographs (**A**) were taken by a stereo microscope. The red curves in (**B**) represent the abscission layer (AL). pa, palea; le, lemma; ra, rachilla; sl, sterile lemma; rg, rudimentary glume; pe, pedicel; vb, vascular bundle; sz, supporting zone. Bars = 1 mm.

**Figure 2 ijms-24-08889-f002:**
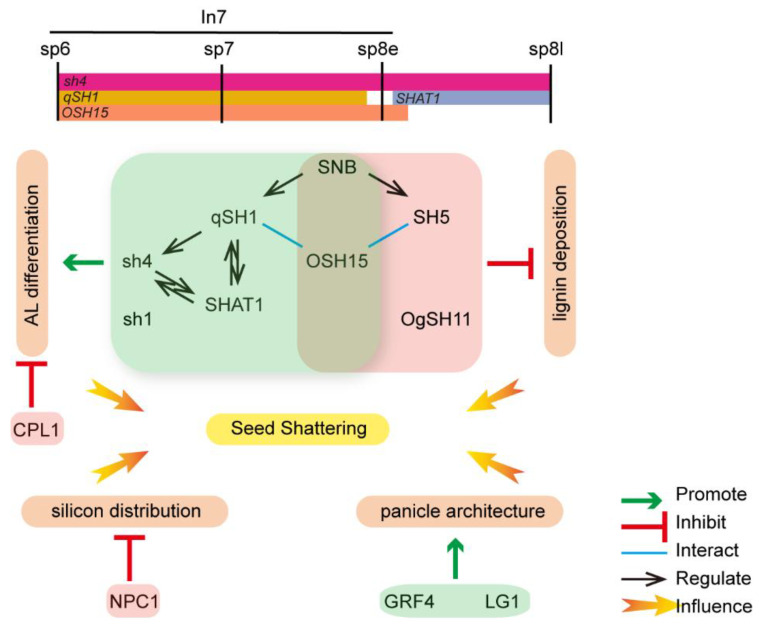
Molecular genetic regulatory network of rice seed shattering. The upper half of this figure shows the expression period of shattering genes during inflorescence and spikelet development. The bottom half of this figure indicates the genetic model of the regulatory network influencing seed shattering.

**Table 1 ijms-24-08889-t001:** Shattering QTLs that have been mapped in rice.

Chr	QTL	Cross Parents	Species	Marker Interval	Ref.
1	*qSH1*	H359 × Acc8558	*Indica* × *Indica*	Xpsr598-C161	[49]
1	*qSH1-1*	Zhenxian97B × Miyang46	*Indica* × *Indica*	RG447-RG472	[40]
1	*Sh-4*	ZYQ8 × JX17	*Indica* × *Japonica*	G3702-RG541	[50]
1	*sh1.1*	*O. rufipogon* × Jerrerson	*O. r.* × *Japonica*	RM315-RG331	[51]
1	*Qss1*	Samgang × Nagdong	*Indica* × *Japonica*	RM6696-RM476	[52]
1	*Qal1*	Samgang × Nagdong	*Indica* × *Japonica*	RM6696-RM476	[52]
1	*Sh-2*	Miara × C6′	*Japonica* × *WR*	RM212	[53]
1	*qSHT-1*	W1944 × Pei-Kuh	*O. r.* × *Indica*	G393-OP	[54]
1	*qWDSH1*	WR04-6 × Qishanzhan	*WR* × *Indica*	-	[55]
1	*sh1*	P16 × Aijiao Nante	*O. r.* × *Indica*	RZ161-DRG470X	[56]
1	*Sh(t)*	NS2098 × GuiNongZhan	*Javanica* × *Indica*	RM3825-RM11875	[39]
1	*qSH-1-1*	93-11 × Nipponbare	*Indica* × *Japonica*	RM472-RM1387	[57]
1	*qWDSH2*	WR04-6 × Qishanzhan	*WR* × *Indica*	-	[55]
2	*qSH2*	H359 × Acc8558	*Indica* × *Indica*	C777-R1843	[49]
2	*qSH2*	Zhenxian97B × Miyang46	*Indica* × *Indica*	RM263-RM240	[40]
2	*qSH2.1*	DP15/DP30 × 9311	*O. r.* × *Indica*	C2-21~C2-22	[58]
3	*Qss3*	Samgang × Nagdong	*Indica* × *Japonica*	RM426-RM570	[52]
3	*qSH3*	SS18-2 × EM93-1	*WR* × *Indica*	RM486	[59]
3	*sh3*	P16 × Aijiao Nante	*O. r.* × *Indica*	RM282-RZ672	[56]
3	*sh3.2*	*O. rufipogon* × Jerrerson	*O. r.* × *Japonica*	Rm282	[51]
3	*sh3.1*	*O. rufipogon* × Jerrerson	*O. r.* × *Japonica*	RM60	[51]
3	*qSH3-1*	Zhenxian97B × Miyang46	*Indica* × *Indica*	RZ448-RZ519	[40]
3	*qSH3-2*	Zhenxian97B × Miyang46	*Indica* × *Indica*	RZ142-RZ613	[40]
4	*Qss4*	Samgang × Nagdong	*Indica* × *Japonica*	4002-4007	[52]
4	*sh4.1*	*O. rufipogon* × Jerrerson	*O. r.* × *Japonica*	RZ656-RM185	[51]
4	*qSH4-1*	H359 × Acc8558	*Indica* × *Indica*	P76/M22-4-C140	[49]
4	*qSH4-2*	H359 × Acc8558	*Indica* × *Indica*	Xpsr150-Xpsr488	[49]
4	*sh4*	Dinalaga × SR-5	*Javanica* × *Indica*	R250	[60]
4	*sh4*	P16 × Aijiao Nante	*O. r.* × *Indica*	RG620-R416	[56]
4	*qSH4*	Zhenxian97B × Miyang46	*Indica* × *Indica*	RM241-RZ675	[40]
4	*qSHT-4*	W1944 × Pei-Kuh	*O. r.* × *Indica*	CDO244-RG214	[54]
4	*qSH4*	SS18-2 × EM93-1	*WR* × *Indica*	RM471	[59]
4	*qSH4.1*	DP15/DP30 × 9311	*O. r.* × *Indica*	C4-22~C4-23	[58]
5	*Qss5-2*	Samgang × Nagdong	*Indica* × *Japonica*	5028-5037	[52]
5	*Qal5-1*	Samgang × Nagdong	*Indica* × *Japonica*	5021-RM289	[52]
5	*qSH5.1*	DP15/DP30 × 9311	*O. r.* × *Indica*	RM3227-5M13153	[58]
6	*qSH6*	Zhenxian97B × Miyang46	*Indica* × *Indica*	RM225-RM197	[40]
6	*SH6(t)*	Nipponbare × R225	*Japonica* × *Indica*	RM253-ZTQ53	[61]
6	*sh6*	P16 × Aijiao Nante	*O. r.* × *Indica*	RG64-R2123	[56]
6	*qSH-6-1*	93-11 × Nipponbare	*Indica* × *Japonica*	RM6782-RM3430	[57]
6	*qsh6*	Cheongcheong × Nagdong	*Indica* × *Japonica*	RM20632–RM439	[62]
6	*qSH-6*	H359 × Acc8558	*Indica* × *Indica*	C488-P22/M17-33	[40]
7	*sh7.1*	*O. rufipogon* × Jerrerson	*O. r.* × *Japonica*	RG30-RM214	[51]
7	*qSH7*	Zhenxian97B × Miyang46	*Indica* × *Indica*	RM18B-RZ989	[40]
7	*qSH7-1*	H359 × Acc8558	*Indica* × *Indica*	P22/M17-25-P19/M76-5	[49]
7	*qSH7-2*	H359 × Acc8558	*Indica* × *Indica*	P76/M122-1-Xpsr130	[49]
7	*qSH7*	SS18-2 × EM93-1	*WR* × *Indica*	RM471	[59]
8	*sh8*	P16 × Aijiao Nante	*O. r.* × *Indica*	RG333-C1121	[56]
8	*sh8.1*	*O. rufipogon* × Jerrerson	*O. r.* × *Japonica*	RM42-RM44	[51]
8	*qSHT-8*	W1944 × Pei-Kuh	*O. r.* × *Indica*	G187-G2132	[54]
8	*qSH8*	SS18-2 × EM93-1	*WR* × *Indica*	RM135B	[59]
8	*Qps8*	Samgang × Nagdong	*Indica* × *Japonica*	8020-8026	[52]
9	*Qbs9*	Samgang × Nagdong	*Indica* × *Japonica*	9004-9008	[52]
9	*qSH9.1*	DP15/DP30 × 9311	*O. r.* × *Indica*	C9-8~C9-9	[58]
10	*qSH10*	Zhenxian97B × Miyang46	*Indica* × *Indica*	RG257-RM258	[40]
11	*qSHT-11*	W1944 × Pei-Kuh	*O. r.* × *Indica*	G24-G320	[54]
11	*sh-5*	ZYQ8 × JX17	*Indica* × *Japonica*	G320-RZ141	[50]
11	*qSH11*	Zhenxian97B × Miyang46	*Indica* × *Indica*	RM187-Z536	[40]
11	*qSH11.1*	DP15/DP30 × 9311	*O. r.* × *Indica*	C11-5~C11-8	[58]
11	*qSH11.2*	DP15/DP30 × 9311	*O. r.* × *Indica*	C11-16~C11-17	[58]
12	*qSH12*	Zhenxian97B × Miyang46	*Indica* × *Indica*	RG81-RM313	[40]

“-” indicates it is not mentioned in the article. “*WR*” means weedy rice; *O.r.* means *O. rufipogon*.

**Table 2 ijms-24-08889-t002:** Cloned genes for seed shattering in rice.

Gene	Genomic Locus	Cross Parents	References
*qSH1*	LOC_Os01g62920	Kasalath × Nipponbare	[24,63]
*sh4*	LOC_Os04g57530	*indica* × *O. nivara*	[13,64]
*CPL1*	LOC_Os07g10690	Hwacheong × Blue & Gundil	[41,65]
*SHAT1*	LOC_Os04g55560	W1943 × Guangluai 4	[66]
*Sh1*	LOC_Os03g44710	*indica cv* Nanjing 11	[67,68]
*SH5*	LOC_Os05g38120	*japonica* cv. Dongjing	[69]
*OSH15*	LOC_Os07g03770	*japonica* cv. Dongjing	[70]
*SNB*	LOC_Os07g13170	*O. rufipogon* × Teqing	[71]
*OgSH11*	XM_052280091 ^※^	Taichung65 *×* WK21	[72]
*NPC1*	LOC_Os03g61130	Zhonghua11	[73]
*GRF4/PT2*	LOC_Os02g47280	R1126 *×* CDL	[74]
*LG1*	LOC_Os04g56170	Nipponbare *×* W630	[75]

“※” indicates NCBI reference sequence.

## Data Availability

No new data were created or analyzed in this study. Data sharing is not applicable to this article.

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
