# Peer review of "Advances in Rice Seed Shattering"

_ijms, 2023, doi:10.3390/ijms24108889_

Round 1

Reviewer 1 Report

This review discusses the physiological basis of rice seed shattering, environmental factors influencing the trait, as well as the main QTLs, genes and regulatory network linked to the trait in two subspecies of Oryza sativa (ssp. indica and japonica) and some other Oryza species (the current state). The authors touch upon the issue of domestication, since the seed shattering is considered one of the key signs of rice artificial domestication.

The manuscript may be accepted for publication in IJMS after a minor revision.

L234: “An increasing number of genes related to abscission layer formation or seed shattering have been identified up to now, but it is not the same as the dicot model plant Arabidopsis thaliana which is well studied…”

The authors briefly mention that the regulation of the trait in rice (monocot) differs from that in the model plant Arabidopsis (dicot). However, it did not sound at all whether rice and other Poacea species have anything in common. The family includes many crops (for example, wheat, rye, barley, maize), at one time, like rice, domesticated by humans, and shattering of seeds was also a key selective trait. Whether or not the evolution and/or domestication of these species are convergent with rice? It would be nice to clarify this issue for the readers (in chapter 6). As a source [doi: 10.1007/s10142-005-0015-y; 10.1016/bs.ctdb.2016.02.002].

Chapter 7 can be supplemented with information on what biotechnological work is already underway to obtain varieties of rice with non-shattering seed pods using molecular genetic knowledge. It is possible to discuss the development strategy of rice biotechnology in this direction using, for example: 10.1007/s11248-017-0016-3; 10.3390/biology11121823; 10.3389/fpls.2020.00619; 10.1371/journal.pone.0229782.

Other remarks:

In the abstract and in the text: “Character” should be replaced with “Trait”.

L141: “Homeobox gene, qSH1 and SH5” – please, replace with “BEL-1 type homeobox transcription factor qSH1 and KNOX protein SH5”

L50 and 177: The abbreviation for abscission zone (AZ) was entered twice. Please, leave only one (at Line 50). The abbreviation of the abscission layer (AL) was introduced once, but the authors almost never use it. The complete phrase occurs a lot of times (see Lines 51, 54, 55, 57, 62, 63, 64, 65, 67, 68 and so on…). Once an abbreviation has been introduced, it is necessary to correct the text and replace (at least in most cases) "abscission layer" with "AL"

English language in the manuscript is quite literate and understandable, but a minor professional editing is still needed (at the discretion of the Editor).

Reviewer 2 Report

Dear the editor,

The submitted review manuscript includes comprehensive references and data on recent research papers, in addition to papers published some decades ago, studying seed shattering in rice.

Basically, I do not have additional scientific information to the manuscript. However, in order to clarify the contents described in the manuscript, I would like to make some comments on a section of QTL and genes.

First, QTLs are identified by a comparison between two parents and may explain the deference in a trait between the parents as genetic reason. In the case of the seed shattering, some QTLs explain inter-species differences and others explain intra-species ones. I think it would be better to write this point clearly in the manuscript. For example, since table 1 contains both QTLs, the table might be changed to be displayed more clearly.

In table1, sometimes “Rufipogon” is used. On the other hand, accession number is also used to cite the strain in rufipogon. It is better to use consistent statement.

qWDSH1 is identified a cross of a weedy rice accession and a cultivated rice. Please add the accession names in table 1.

In figure2, all genes are displayed in one figure. Although I believe that genes involved in regulation of AZ and seed shattering are most likely common among O. sativa, O.glaberrima and their ancestors, it is still under working to elucidate the genetic mechanisms in Oryza species since they are distinct species. From this view point, I am not sure that we should make one picture with mixing the studies on different species.

Lastly, AL in Figure1 seems to separate only sterile lemmas. There is a stem like tissue between sterile lemmas and rudimentary glumes, and AL is formed in the tissue.

Reviewer 3 Report

The chief weakness of this review is that it is a mere compilation of results from many papers that does not provide a conceptual framework to integrate and critically analyze the referred mass of information. It seems the Introduction of a student’s thesis (a good one, anyway). Although a list of studies focalizing on the present topic can be useful by itself, a review must also extract from the many individual results some general inference, or diverse inferences, one for each sub-topic. Otherwise, after reading the review a reader remains with a question: “So, what”? This was, at least, my reaction. Even a review paper should provide some novelty to be published on a good scientific journal, in my opinion.

Minor remarks are detailed below.

Line 10, “natural environment”: seed shattering is also an important character for weedy rice to compete with the rice crop.

Lines 13-14, “shattering character is the key character selected in rice breeding”: after domestication, rice breeding has focused on several key characters, taking the non-shattering trait as a given, since breeding is based on crosses between already non-shattering cultivars.

Line 30: “symbol”? Perhaps, ‘epitome’?

Throughout the text, the Authors frequently use the past tense (e.g., “varied”, line 33). This is appropriate when one refers to a specific experiment that has been concluded. In general statements about the present state of the art, the present tense is commonly used.

Throughout the text, particularly in the Introduction section, statements about the state of the art of the matter require one or more references.

Line 52: “arrangement”?

Lines 52-554 and Fig. 1: this is wrong. Glumes are a tiny part of the hull, in rice. The present authors should look at Li et al. (2018. Control of grain size in rice. Plant Reproduction 31:237-251) to learn some basic information about the rice grain.

Lines 56-57, “when gametophyte cells began to differentiate in young inflorescence”: when gametophyte cells began to differentiate, the inflorescence has become an infructescence (namely, a panicle).

Line 63: what is the “intercellular layer”? Do the Authors mean the middle lamella, which is a layer, made up of calcium and magnesium pectates, that cements together the primary cell walls of two adjoining plant cells? As for several other statements, a suitable reference is missing.

Line 64: what is “trans-differentiation”?

Lines 88-89, “In addition to the development of abscission layer, many environmental factors can also lead to rice shattering”: don’t the environmental factors lead to rice shattering at the abscission layer? Please, re-word this sentence.

Lines 102-103 replicates what said on lines 58-59.

Line 124: “traits is”?

Lines 125-128, “Another viewpoint is that rice shattering is a quality trait and is mainly regulated by a single gene [32]. Up to 2020, at least 60 QTLs/genes related to rice shattering were detected in 12 rice chromosomes using various genetic populations and linkage maps (Table 1)”: the latter sentence disproves the former.

Line 130: please, change “alleles with” to “allelic with”. The same for line 131.

Lines 136-142: the differences between these gene lists are not entirely clear.

Line 154, “Lin, et al.”: I suppose the comma is superfluous. The same holds true for several other instances when Authors are quoted in the text.

Line 237: “… at BASE 237 …”. Or ‘nucleotide site’.

Line 165, “REPLUMLESS (RPL)”: italics. The same for SH5 on line 170; and for qSH3 on lines 199 and 202. Also, SPR3 on line 225; and qSH1 on line 302.

Line 191: “controls”.

Line 231: “IS associated”.

Line 232, “No research has been reported previously to be related with rice seed shattering”: for the previously mentioned gene only, I suppose.

Lines 234-237: not clear.

Lines 239-240 and Fig. 2: who proposed this model? Is the figure reproduced with permission? As already said, many statements are provided without a reference. This is not acceptable in general, and surely not in a review paper.

What are sp6 to sp8 in Fig. 2? They are mentioned as ‘stages’ on lines 249-255, but also In7 is called a ‘stage’ (line 252), though it is represented as a hierarchically higher thing in Fig. 2. This is incomprehensible.

Line 262: “homologous”.

Line 278, “Simultaneous tillering and ripening”: in cereals? Is this an achievement of domestication? Who said this? Try using a combine in a paddy where rice is both ripen and has also produced many new fertile tillers: it is a mess.

Line 279-280, “the comprehensive domestication characteristics of rice (Oryza sativa) are very obvious, involving over 20 agronomic traits”: I am not able to make a list of all these 20 agronomic traits that are very obvious domestication characteristics of rice, unless I carefully look through the literature on the topic. Hence, they are not all so obvious.

Line 285, “than”: ‘more than’ or ‘less than’?

The Conclusions are largely about “breeding applications”. However, “Although a number of QTLs have been identified to control shattering, only a limited number of genes have been cloned, and how other genes affect shattering remains unclear. Moreover, Rice seed shattering is a complex multifaceted regulation process” (lines 330-332). Since rice seed shattering is a complex multifaceted regulation process that is not fully understood, how can it be quickly used for “breeding applications”? Apart from some specialized studies, I am afraid ordinary rice breeding programs are not yet using molecular markers to control shattering. Even because, as said, breeding is largely based on crosses between already non-shattering cultivars. The present review, in fact, does not suggest any breeding strategy to obtain the moderately shattering rice required by combine harvesters (lines 38-40) from crosses wherein strong shattering progenies occur. Usually, such crosses are performed to introduce in a good cultivar some novel trait from non-elite materials.

Although a list of publications is a useful and necessary starting point for a review, much more must be elaborated from that in a review paper. An integrated view conveying some novel information is an important feature readers look for in a good review.

Minor editing of English language is required.

Round 2

Reviewer 3 Report

The Authors have provided extensive revisions and improvements to the manuscript. The new text requires editing of English language, however.

Anyway, I still think that this Review does not provide a conceptual framework to integrate and critically analyze the referred mass of information that is novel, or structured enough, to be published on the IJMS in the present form. Specifically, I think that "the relationship between seed-shattering genes and domestication" (lines 19-20) is poorly discussed in this Review, even though several good Reviews on rice domestication are present in the literature. In this regard, stating that "Many genes related to shattering in Gramineae plants (Poaceae) were found to be the homologous genes of rice, indicating a possibility of convergent evolution in shattering between them" (lines 366-368) is a demonstration that the present Authors do not understand what they are speaking of: "convergent evolution" is convergence of phenotype; instead, gene homology within a taxonomic family is due to the phylogenetic origin of the species of that family from a common ancestor, from which they inherited that gene. This is basic knowledge of inheritance.

What "allotetraploid rice" (line 399) is must be introduced prior to making bold statements about "a new grain to enhance world food security".

As the Authors just mentioned gene editing "in a weedy rice line", it is not clear how "the seed shattering ability and dispersibility of weedy rice are greatly reduced by editing seed shattering genes to control its population spread and reduce the harm of weedy rice" (lines 410-412). It would seem that the Authors are proposing to artificially create new weedy rice lines that do not shatter, with the aim to reduce the spread of weedy rice. This would, in fact, cause the almost immediate disappearance of these artificial lines, whereas the normal weedy rice populations would continue to spread as before. Perhaps, the Authors are speaking of editing the shattering genes of new rice crop varieties so that the improved genes carried from these new rice crop varieties (maybe resistance genes) are less probable to spread to weedy rice? Unfortunately, this is not what the Authors have written. In addition, recombination of the edited shattering genes and the resistance genes would ultimately lead to stronger weedy rices, albeit after an initial noticeable lag time.

Apart from the novel allotetraploid rice and new rice crop varieties with partially silenced sh4 expression, the present review does not suggest any breeding strategy to obtain the moderately shattering rice required by combine harvesters mentioned on lines 40-47.

As the present Authors call "sterile lemma" what Li et al. (2018. Control of grain size in rice. Plant Reproduction 31:237-251) call 'glumes', can the present Authors quote a suitable reference? I'd suggest Yoshida and Nagato (2011. Flower development in rice. J. Exp. Bot. 62:4719-4730).

Minor editing of English language is required. In particular, for the text added with the last revision.

Round 3

Reviewer 3 Report

In the last version of their manuscript, the Authors have provided extensive revisions and improvements to the manuscript, complying with the remarks I’ve previously raised. Now I think that the present Review provides interesting information structured enough to be published on the IJMS. The conceptual framework used by the present Authors in this Review differs in several points from what I personally think about this matter, but this is, in fact, a matter of opinion.

I only wish to point out that the fact that “As with rice, reducing seed shattering in other cereal crops (Poaceae) is one of the most significant changes during domestication, indicating a possibility of convergent evolution in shattering between them” (lines 373-376) is obviously due to human selection acting upon any cereal crop to minimize seed shattering before harvest. This is clearly stated in several reviews on this topic, both focused on rice as well as on crops in general. This ought to be stated clearly in the manuscript.
